SOFTWARE

# TARPON—A Telomere Analysis and Research Pipeline Optimized for Nanopore

**Nathaniel Deimler** [1,2*], **David V. Ho** [1,2], **Norbert Paul** [3], **Zoë Gill** [1,2], **Peter Baumann** [1,2,4*]

**1** Department of Biology, Johannes Gutenberg University, Mainz, Germany, **2** Institute for Quantitative and Computational Biosciences, Mainz, Germany, **3** Institute for the History, Philosophy, and Ethics of Medicine, Johannes Gutenberg University Medical Center, Mainz Germany, **4** Institute of Molecular Biology, Mainz, Germany

\* nathanieldeimler.research@gmail.com (ND); peter@baumannlab.org (PB)

**Data availability statement:** Putative telomeric sequences containing at least ten non-consecutive telomeric repeats are provided for the four samples described in this study and are publicly available on SRA under BioProject #PRJNA1313423. TARPON is publicly available at https://github.com/baumannlab/TARPON.

## Abstract

Long-read sequencing has transformed many areas of biology and holds significant promise for telomere research by enabling analysis of nucleotide-level resolution chromosome arm–specific telomere length in both model organisms and humans. However, the adoption of new technologies, particularly in clinical or diagnostic contexts, requires careful validation to recognize potential technical and computational limitations. We present **TARPON** (Telomere Analysis and Research Pipeline Optimized for Nanopore), a best-practices Nextflow pipeline designed for the analysis of telomeres sequenced on the Oxford Nanopore Technologies (ONT) platform. TARPON can be executed via the command line or integrated into ONT's EPI2ME agent, providing a user-friendly graphical interface for those without computational training. Nextflow's container-based architecture eliminates dependency conflicts, thereby streamlining deployment across platforms. TARPON isolates telomeric repeat–containing reads, assigns strand specificity, and identifies enrichment probes that can be used both for demultiplexing and for confirming capture-based library preparation. To ensure that the analysis is restricted to full-length telomeres, reads lacking a capture probe or non-telomeric sequence on the opposite end are excluded. A sliding-window approach defines the subtelomere-to-telomere boundary, followed by quality filtering to remove low-quality or subtelomeric reads that passed earlier steps. The pipeline generates customizable statistics, text-based summaries, and publication-ready visualizations (HTML, PNG, PDF). While default settings are optimized for diagnostic workflows, all parameters are easily adjustable via the GUI or command line to support diverse applications. These include telomere analyses in variant-rich samples (e.g., ALT-positive tumors) and organisms with non-canonical telomeric repeats such as some insects (GTTAG) and certain plants (GGTTTAG). TARPON is the first complete and experimentally validated pipeline for Nanopore-based telomere analysis requiring no data pre-processing or prior bioinformatics expertise, while offering flexibility for advanced users.

**Funding:** This work was funded in part by an Alexander von Humboldt Professorship awarded to P.B. at JGU. The funders had no role in study design, data collection or analysis, decision to publish, or preparation of the manuscript.

**Competing interests:** The authors have declared that no competing interests exist.

## Introduction

Telomeres, the structures that protect the ends of linear eukaryotic chromosomes, are comprised of G-rich DNA repeat sequences, proteins, and telomeric RNA [1]. In humans, telomeres span from 3 to 15 kbp and terminate in a 100–200 nucleotide single-stranded G-rich 3′ overhang [2,3]. Each time a cell duplicates, approximately 50–150 base pairs of terminal sequence are lost from each chromosome end due to the end replication problem [4,5], resulting in progressive telomere shortening [6,7]. Telomeric repeat arrays protect the integrity of the coding regions of the genome by temporarily buffering this sequence attrition. However, in the absence of mechanisms that replenish telomeric DNA, progressive telomere shortening eventually triggers cellular senescence as telomeres reach a critical length [8]. Telomere length is maintained in highly proliferative cells by telomerase, a ribonucleoprotein complex comprised at its core of the catalytic protein Telomerase Reverse Transcriptase (TERT) and a non-coding Telomerase RNA subunit, TR [9,10]. Telomere Biology Disorders (TBDs) are a symptomatically heterogeneous group of syndromes associated with abnormally short telomeres or telomere instability [11,12]. The broad spectrum of symptoms resembles the aging process, ranging from changes in skin pigmentation and nail dystrophy to severe effects such as pulmonary fibrosis and total bone marrow failure [13].

Accurate measurement of telomere length is therefore essential for both research and clinical applications. Multiple techniques have been developed including, but not limited to, terminal restriction fragment (TRF) analysis by Southern blotting, quantitative PCR (qPCR), and fluorescent in situ hybridization (FISH) [14]. Terminal restriction fragment (TRF) length analysis has long been considered the gold standard in research settings [15,16]. In this method, genomic DNA is digested by restriction enzymes that frequently cut within the genome but not within telomeric repeats. Digested DNA is then resolved on an agarose gel and hybridized with a labeled telomere-specific probe. Although this method shows low inter-laboratory variation [17], reproducibility is limited by poorly described protocols for digestion and gel quantification [18]. Moreover, variations in subtelomeric sequences or DNA modifications may lead to apparent inter-individual differences in telomere length [19].

qPCR estimates telomere length indirectly by comparing the quantity of telomeric repeat amplification products to a single-copy gene product [20]. Its simplicity, low cost, and minimal DNA input requirements have enabled widespread use in clinical samples and large-scale comparisons [21,22]. However, qPCR has been shown to yield variable results depending on laboratory protocols [17], DNA extraction methods [23], and storage conditions. For instance, samples stored in 4% formaldehyde showed increased telomere length measurements over time, a bias not observed with samples preserved in RNAlater [24].

Flow-FISH, currently the clinical gold standard, uses fluorescently labeled peptide nucleic acid (PNA) probes to detect telomeric DNA in permeabilized cells, with fluorescence intensity measured by flow cytometry [25]. It avoids certain biases associated with TRF and qPCR and enables direct comparison to bovine reference standards [26]. However, Flow-FISH requires fresh blood samples, limiting its use on archived or bio-banked material.

In summary, each of the established methods for telomere length measurement has specific strengths and limitations. A reliable, reproducible, and user-friendly method that works across a wide range of sample types—including fresh and archived specimens—remains a critical but unmet need.

Third-generation long-read sequencing has emerged as a powerful tool across biological disciplines, including aging and telomere research. Oxford Nanopore Technologies (ONT) sequencing has enabled nucleotide-resolution analysis of human telomeres [27–29]. In parallel, PacBio HiFi sequencing has also been used to generate high-throughput, single-molecule telomere length measurements at nucleotide resolution across diverse human cell lines and patient samples [30]. However, telomeres represent only ~0.01% of the human genome, and whole-genome sequencing yields relatively few telomeric reads. To address this, two enrichment strategies have been developed: a physical enrichment using biotinylated duplexes and streptavidin-coated beads ("duplex-enriched") [29] and a library preparation method that captures telomeric ends using a telomere-specific splint and ONT's adapter overhang ("splint-enriched") [27,28]. Both strategies increase telomeric read recovery and append a known capture probe to the distal telomere end, confirming full-length capture.

Unlike traditional approaches that return a single statistic (usually mean or median) that represents the telomere length per sample, long-read sequencing enables single-molecule resolution of telomere length distributions. With sufficient subtelomeric sequence, reads can be aligned to the genome, allowing chromosome arm-specific telomere assignment, as shown in yeast [31] and the human cell line HG002 [27–29]. However, this approach is currently limited to samples with high-quality subtelomeric reference assemblies [32].

Despite these promising advances, no standardized and validated pipeline exists for the analysis of Nanopore-based telomere data. Here, we present **TARPON** (Telomere Analysis and Research Pipeline Optimized for Nanopore), an experimentally validated, end-to-end pipeline for analyzing telomere reads obtained via ONT sequencing. TARPON is implemented in Nextflow and can be executed via the command line or through the user-friendly EPI2ME graphical interface, which requires no programming experience. The pipeline supports both duplex- and splint-enriched libraries and includes automated quality control, capture probe detection, telomere length quantification, and comprehensive reporting. Parameters can be modified through either interface, enabling both novice and expert users to tailor TARPON to a wide range of experimental applications.

## Design and implementation

### Ethics statement

The Clinical Ethics Committee of the Johannes Gutenberg University Medical Center, acting in its capacity as an Institutional Review Board (IRB), has reviewed the research project and determined that all human genomic material used in the study is fully anonymized with no means of re-identification, written informed consent for research use has been obtained for all human-derived samples, including secondary use, and the public datasets are used in accordance with their respective data use agreements. Given that these conditions are implemented, no formal ethical approval is required, and the Committee does not object to the continuation or publication of the study.

TARPON addresses the computational challenges associated with telomere sequencing using Oxford Nanopore long-read technologies by providing an integrated analysis pipeline suitable for bioinformaticians, researchers, and clinicians. The pipeline requires no preprocessing of the data and accepts a range of input formats, including FASTQ, compressed FASTQ, and BAM files. It supports data that has been basecalled using any of ONT's fast basecalling models as well as the super-high accuracy (SUP) model, simplifying usage and eliminating the need for manual data manipulation prior to analysis.

Once provided with input files, TARPON identifies putative telomeric reads, assigns strand specificity, detects the subtelomere-to-telomere boundary, and applies several filtering steps. These include the removal of reads lacking a terminal capture probe added during telomere enrichment (see sections b and c in S1 Methods for more information), reads dominated by telomeric repeats at the 5′ end (subtelomeric end), reads with extended regions of erroneous repeat

calls, and reads in which the telomere start site is misidentified. The pipeline generates multiple files summarizing each processing step, including telomere read counts, read-level length and quality statistics, as well as bulk telomere length distributions, returned in both tabular and graphical formats. If the sequencing run is multiplexed, TARPON automatically separates all statistics and plots in a sample-specific manner. In addition to the generation of raw files (PDF, PNG, and TXT), TARPON compiles all relevant information and figures into an HTML report (Fig 1a).

For the more experienced user, TARPON can be cloned directly from GitHub and run from the command line after installation of Java, Nextflow, and Docker. No installation of additional software dependencies is required, as TARPON utilizes Docker containers through Nextflow, inherently avoiding version incompatibility issues.

For users less familiar with the command line, TARPON is integrated into ONT EPI2ME agent. This platform assists with the installation of Java, Docker, and Nextflow on a local machine, enabling GUI-based operation of the pipeline. Integration into EPI2ME is achieved by entering the GitHub repository URL into the EPI2ME "Add Workflows" utility. Once loaded, the user can specify pipeline inputs and adjust parameters either through the GUI or the command line. Configurable options include the capture probe sequence, barcode files for sample demultiplexing, and parameter settings for telomeric read isolation, capture probe and barcode detection, subtelomere-to-telomere boundary identification, and the stringency of each filtering step. Additional guidance on all configurable parameters is available in the README file provided with the pipeline. TARPON is publicly available at https://github.com/baumannlab/TARPON.

## Results

Information on samples used in this study, telomere enrichment methodology, basecalling, and TARPON pipeline execution can be found in the attached S1 Methods.

### First pass telomeric read isolation

Regardless of which telomere enrichment technique is used, the low abundance of telomeric DNA relative to bulk genomic DNA results in a high proportion of non-telomeric reads in the sequencing output. Performing telomere-specific functions, such as identifying the subtelomere-to-telomere boundary, on the full dataset would unnecessarily increase computational demands and analysis time. TARPON addresses this by first isolating putative telomere-containing reads based on the presence of a user-defined telomeric repeat motif, which defaults to the canonical vertebrate repeat GGTTAG.

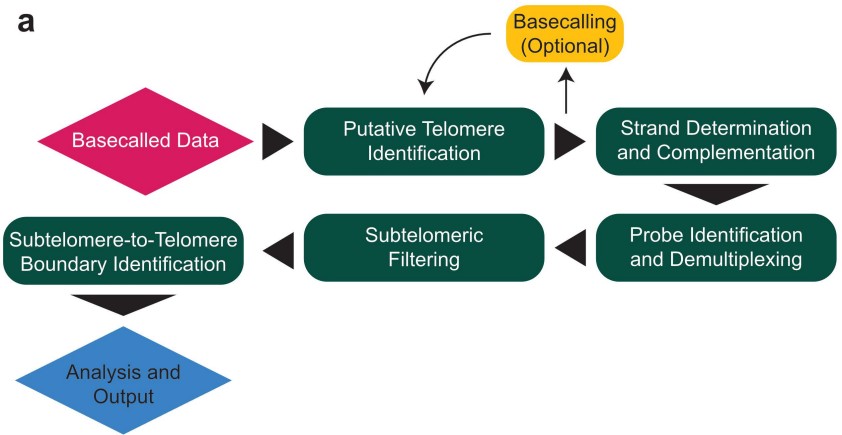

**Fig 1. Pipeline processes. (a)** Graphical display of the pipeline workflow and strategy.

To establish an efficient and consistent strategy for first-pass telomeric read isolation, we tested multiple parameter combinations across two splint-enriched sequencing runs (HG002-SE and WB60-SE) and two duplex-enriched runs (HEK-DE and WB60-DE). Raw pod5 files, which store the original electrical signal data generated by the nanopore sensor and serve as the source for all downstream read information, were basecalled using dorado v0.7.0 with fast, high accuracy (HAC), and super high accuracy (SUP) models. Our goal was to identify parameters that would yield the same subset of telomeric reads regardless of basecalling model, while minimizing computational overhead even when input file size increased up to 20-fold.

Among the tested strategies, requiring a read to contain at least ten non-consecutive instances of the telomeric repeat (default: GGTTAG) consistently identified the greatest number of reads across all runs (Fig 2a, left). Importantly, this criterion was robust as it produced near-identical read sets across all three basecalling models (Fig 2b). In contrast, increasing the threshold to 20 repeats or requiring repeats to be consecutive led to decreased sensitivity in fast and HAC data relative to SUP, whether detected using custom Python scripts or seqkit grep. These results demonstrate that the 10 non-consecutive repeat threshold provides a basecalling model–agnostic balance between sensitivity and consistency.

Runtime performance for read isolation was also evaluated. Although basecalling model had minimal impact on isolation time, the choice of detection method did: seqkit grep was slightly slower than custom Python scripts. Additionally, identifying non-consecutive repeats took longer than consecutive ones. However, input file size had the greatest influence on runtime. For instance, HEK-DE (2.9 million reads) required substantially more processing time than WB60-DE (1.8 million reads), while both HG002-SE and WB60-SE had under one million reads. Across all strategies, processing time increased linearly with input file size (Fig 2a, lower right). At 20× input size, seqkit grep became significantly slower than custom scripts, likely due to the higher proportion of small, non-telomeric reads in WB60-SE, HEK-DE, and WB60-DE.

To further confirm that reads identified using the ten non-consecutive telomeric repeat criterion originated from the same raw signal data, we compared read IDs across basecalling models for each of the sequencing runs. This analysis demonstrated that the same subset of reads was isolated, regardless of whether fast, HAC, or SUP basecalling was used (Fig 2b). Because the telomeric reads identified in fast and SUP basecalled datasets are identical, TARPON allows users to isolate candidate telomeric reads from fast basecalled data and then selectively re-basecall only these reads using the super high accuracy model. This approach can result in substantial savings in computational resources and user time, since SUP basecalling is performed on only ~1%–2% of the dataset (Fig 2c). To enable this functionality, the user must specify the location of the pod5 files using the --pod5_directory flag and indicate that the data were initially basecalled using a fast model by including the --fast_basecalled parameter.

## Strand orientation of isolated telomeric sequences

Depending on the pre-sequencing enrichment technique, users may expect to sequence either telomeric C-strands alone (e.g., HG002-SE and WB60-SE, Fig 2d) or a combination of both C- and G-strands (e.g., HEK-DE and WB60-DE, Fig 2d). For C-strand-specific splint-enriched sequencing, any read with more than 20% G-strand telomeric repeats is removed from the analysis. In these cases, the --c_strand_only parameter should be set to ensure proper filtering. For duplex-enriched sequencing, where both strand orientations are expected, reads with mixed C- and G-strand identity, defined as containing between 20% and 80% G-strand repeats, are excluded. These are likely chimeric reads or sequences containing only subtelomeric regions and no telomere, but met the 10-repeat threshold.

Telomeric repeats within the subtelomeric portion of a telomere containing read do not affect this filtering step due to their relatively low abundance compared to the telomeric region. Reads classified as C-strand telomeric reads (<20% G-strand content) are reverse-complemented into G-strand orientation for downstream analysis. The original strand identity is retained as a tag in the resulting BAM file to permit future discrimination. This initial filtering step removes approximately 1%–5% of reads, independent of enrichment method (Fig 2e). Only reads passing this strand-filtering step are used in subsequent analysis.

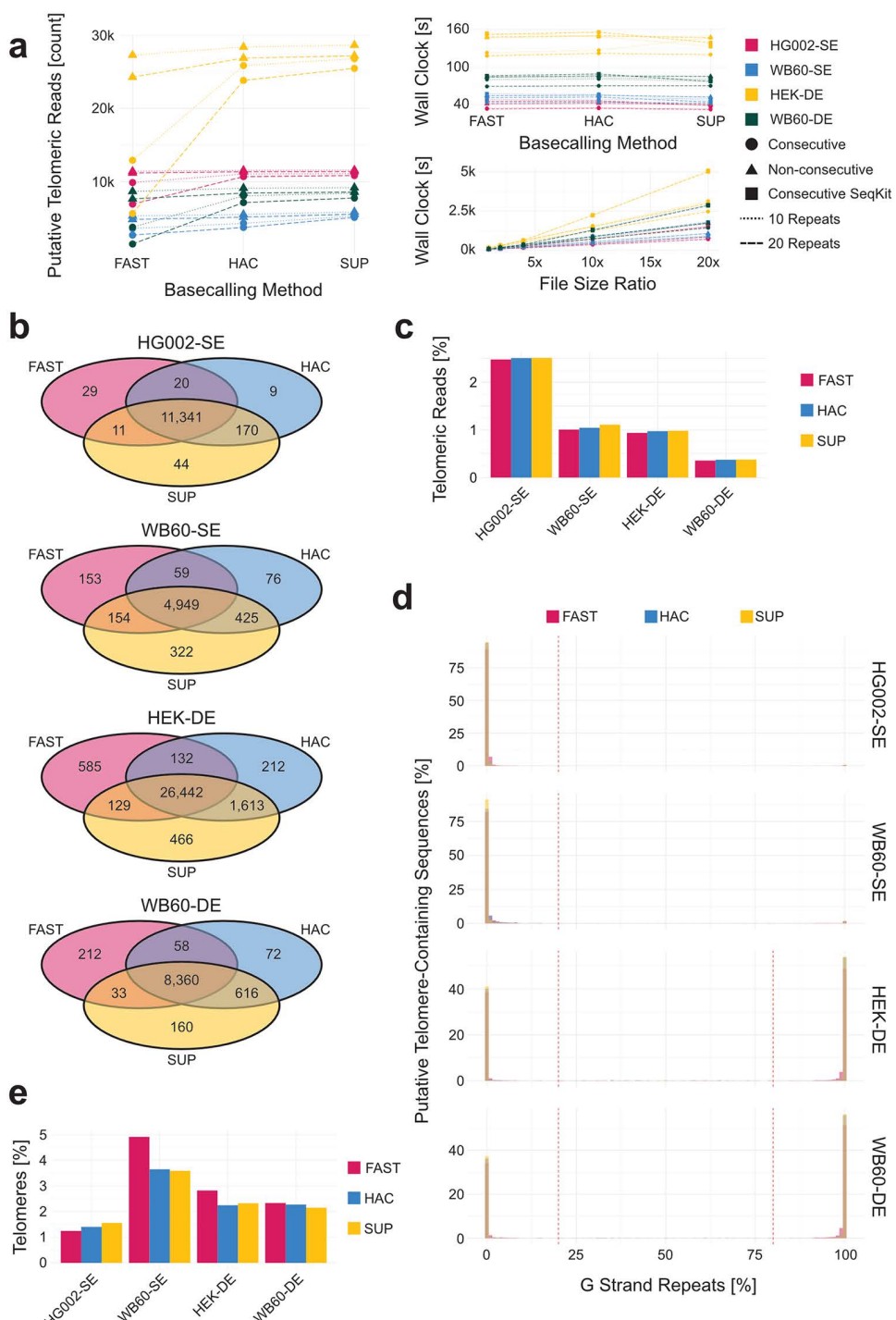

**Fig 2. Putative read isolation and chimera filtering. (a)** Number of putative telomeric reads isolated using fast, high accuracy (HAC), or super high accuracy (SUP) basecalling models across the different isolation methods (left), the speed at which these methods perform (top right), and the speed at which these methods perform on different file sizes (bottom right). **(b)** Overlapping read ids of first-pass isolated telomeric sequences for four sequencing runs using ten non-consecutive telomeric repeats. **(c)** Percentage of reads in a sequencing run that are identified using ten non-consecutive telomeric repeats. **(d)** Percentage of telomeric repeats identified in a read that are G-rich repeats for four sequencing runs across different basecalling models. **(e)** Percentage of telomeric reads removed for containing G-strand repeats when a C-strand enrichment technique is used or chimeric reads when a duplex capture is used.

## Identification of the telomeric capture probe

The strategy of capturing telomeres via ligation of oligonucleotides or partial duplexes to the 5′ end of the C-strand or 3′ end of the G-strand has its roots in earlier ligation-based assays such as the amplification and sequencing of single telomeres [33–35]. Building on this foundation, more recent sequencing-based protocols have refined the technique by integrating capture probes into high-throughput library preparation workflows [27–29].

While the primary purpose of the capture probe is to facilitate enrichment of telomeric DNA fragments, it also serves as a terminal tag that can be identified computationally. This enables verification that a full-length telomere is present within a read and helps exclude fragments that were truncated during library preparation or sequencing. Furthermore, capture probe identification prevents inclusion of extraneous sequences—such as ONT adapter elements or ligation artifacts—in telomere length measurements.

Due to the relatively high error rate of single-pass nanopore sequencing, exact matching of the 12-nucleotide capture probe fails to detect the intended sequence in 20%–50% of reads (Fig 3a, pink bars). Therefore, unless stated otherwise, all results originate from SUP basecalled reads. To improve capture probe identification sensitivity, we tested the effect of permitting a limited number of mismatches. Allowing six errors led to detection of seven or more putative probes per read, an unrealistic outcome that reflects extensive off-target matching (Fig 3a, light green; Fig 3b). Allowing two mismatches resulted in the highest proportion of reads with a single identifiable capture probe across all sequencing runs (Fig 3a, yellow bars), consistent with biological expectations. Increasing the mismatch threshold to three resulted in additional off-target detections (Fig 3a) and a notable increase in runtime (Fig 3b).

Greater than 75% of splint-enriched telomeric reads contained a capture probe when two mismatches were allowed within the 12-nucleotide probe sequence. In contrast, only ~50% of duplex-enriched reads contained a detectable probe under the same criteria. This reduction is likely due to two factors. First, during duplex-enrichment, only one strand of each DNA fragment needs to carry a capture probe for successful streptavidin pulldown, potentially resulting in half of the resulting telomeric reads lacking a probe. Second, only the reverse complement of the GGTTAG permutation was used as a capture probe, which anneals in register with the 5′ end of the C-strand at the double- to single-strand junction [35]. In contrast, the G-strand overhang is more heterogeneous in terminal sequence, and the use of all six possible telomeric repeat permutations may increase probe ligation to G-strand ends. This adjustment may be particularly important when capturing G-strands from telomerase-negative cells where bias for a specific 3′ end permutation is minimal [35]. In duplex-enriched datasets such as HEK-DE and WB60-DE, approximately 30% of G-strand sequences contain a capture probe, compared to 75% of C-strand sequences (S1 Fig). Additionally, capture probe detection is improved by SUP basecalling (Fig 3c). For this reason, we strongly recommend either running TARPON on fast basecalled data with the pod5 directory specified for selective re-basecalling or using pre-basecalled SUP data as input.

Increasing the number of allowed mismatches when identifying the capture probe also increases the number of off-target sequences detected. While the majority of probe matches are located within the final 200 bp of the read (or the first 200 bp in C-strand reads), a small number of matches appear between 4 and 5 kb from the end of the read (Fig 3d). These likely reflect subtelomeric sequences with similarity to the capture probe rather than true probe ligation sites, and represent computational artifacts introduced by relaxed stringency. To mitigate this, an additional filter was introduced to ensure that the capture probe is only accepted if it is the first match found after the identification of twenty telomeric repeats. This positional constraint improves specificity by eliminating internal subtelomeric matches. In splint-enriched datasets, the number of capture probes identified slightly exceeds the number of reads due to abnormal ligation products in which multiple capture probes are present in tandem. This phenomenon is not observed in duplex-enriched libraries, where a substantial fraction of telomeric reads lack a capture probe entirely. Nonetheless, a small number of duplex reads do contain two adapter sequences, as seen in Fig 3a.

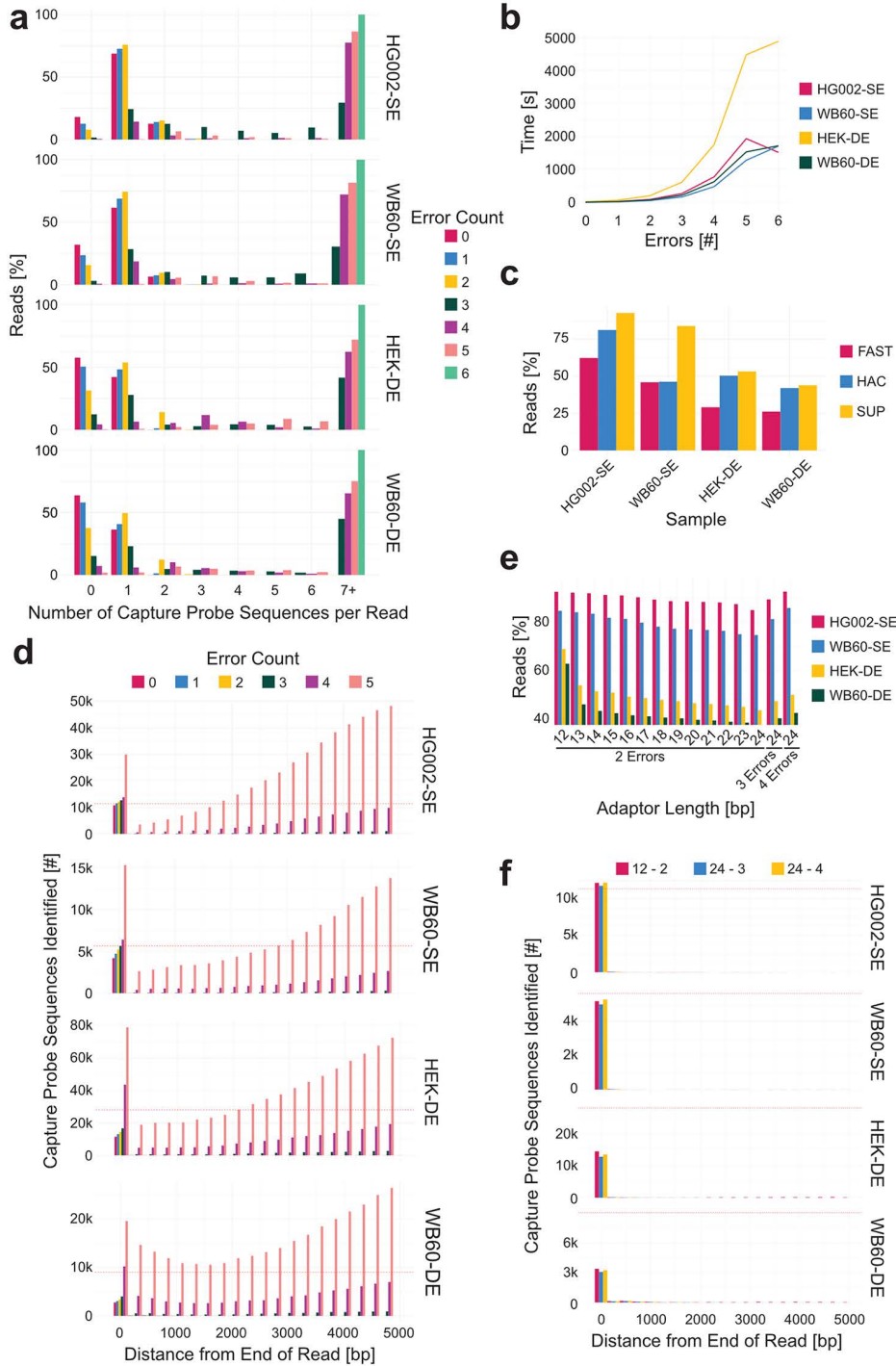

**Fig 3. Identifying the end of a telomeric sequence using an enrichment technique specific capture probe. (a)** Number of capture probe sequences found in SUP basecalled reads while increasing the number of allowed errors within the capture probe sequence. **(b)** Change in wall-clock time associated with increasing the number of allowed errors within the capture probe sequence. **(c)** Percentage of reads in which a capture probe sequence is successfully found using a 12-nucleotide capture probe sequence allowing for two errors. **(d)** Distance between the identified capture probe sequence and the end of the sequence in 200 bp bins. **(e)** Percentage of reads where the capture probe sequence was successfully identified as the capture probe sequence increases in length. **(f)** Distance between the capture probe sequence and the end of the sequence in 200 bp bins while increasing the length of the capture probe sequence and number of allowed errors.

When using capture probe sequences longer than 12 nucleotides, a higher mismatch allowance is recommended to maintain sensitivity. As probe length increases from 12 to 24 nucleotides, retaining only two allowed mismatches leads to a drop in detection efficiency, with fewer reads identified as containing a capture probe (Fig 3e). However, this effect is mitigated by increasing the mismatch threshold. For example, allowing three or four mismatches in 24-nucleotide probes restores capture probe detection to expected levels (Fig 3e). Importantly, increasing both the length of the capture probe and the number of tolerated errors does not result in additional off-target matches within telomeric or subtelomeric regions (Fig 3f), indicating that probe specificity is preserved under these conditions.

## Subtelomere-to-telomere boundary identification

The subtelomere-to-telomere boundary in humans has remained poorly characterized, largely due to the repetitive nature of this region and the historical lack of sequencing technologies capable of resolving it at high resolution. Nanopore sequencing now offers the opportunity to define this boundary on a single-read basis and across chromosome arms. Despite this potential, there is currently no consensus in the field on how to delineate the subtelomere-to-telomere transition. Three recent studies have used distinct methodologies to estimate telomere length from long-read data, complicating cross-study comparisons [27–29].

To develop a consistent and biologically informed algorithm for telomere start detection, we first visualized telomeric sequences across thousands of reads. We observed that the frequency of canonical telomeric repeats (GGTTAG) alone did not consistently define a clear boundary when measured in 100 bp sliding windows (Fig 4a, blue lines). In contrast, a strong and persistent increase in one-nucleotide substitutions of GGTTAG emerged in many reads (Fig 4a, orange lines). Once these variant-containing windows surpassed a certain threshold, they typically maintained >50% signal density across the remainder of the telomere. We refer to this combined pattern of canonical repeats and single-nucleotide substitution repeats as "telomere+1N" and does include insertions or deletion of the wild type repeat. The region enriched for the variants is referred to as the variant repeat-rich (VRR) region and varied in length from a few hundred base pairs (Fig 4a, left) to several kilobases (Fig 4a, middle and right) before transitioning into invariant GGTTAG repeats. As this telomere repeating containing region may arguably have functional relevance, we have included it in the telomere length output. This decision is supported by observations of reads that terminate in a capture probe specific to the enrichment methodology but essentially devoid of a terminal stretch of invariant GGTTAG repeats as seen in Fig 4a (S2a Fig, blue line). Accordingly, TARPON calculates the telomere length from the start of the VRR-region to the distal end of the telomere, as defined by the position of the capture probe.

The telomere length metric requires the ability to accurately detect the start of the VRR-region and the associated frequency spike of telomere+1N repeats. However, while plotting the percentage of telomere+1N repeats within the first 300 bp of a read (the subtelomere end of the read), a notable bimodal distribution emerged: most reads contained less than 15% telomere+1N repeats, while a smaller subset exceeded 80% (Fig 4b). This latter group represents telomere-containing sequences that begin within the VRR-region or even within homogenous GGTTAG repeat sequence. These reads were excluded from further analysis as proximally truncated, since the full length of the VRR-region or telomere sequence cannot be determined with confidence (for C-strand reads, this uncertainty applies to the read end since all reads have been reverse complemented into the G-strand orientation). To avoid underestimating telomere length distributions by including these truncated telomeric reads, it was necessary to exclude 15%–50% of sequences from further analysis (Fig 4c). This percentage was at the higher end of the range in duplex-enriched datasets, likely due to an abundance of short reads corresponding to partial C-strand sequences (S3a Fig), which may have arisen during library preparation and fill-in synthesis of the G-strand overhang.

To validate candidate algorithms for identifying the start of the VRR-region, 100 reads from each sample were selected to represent a variety of subtelomere-to-telomere boundary patterns. These 400 reads were manually annotated to create

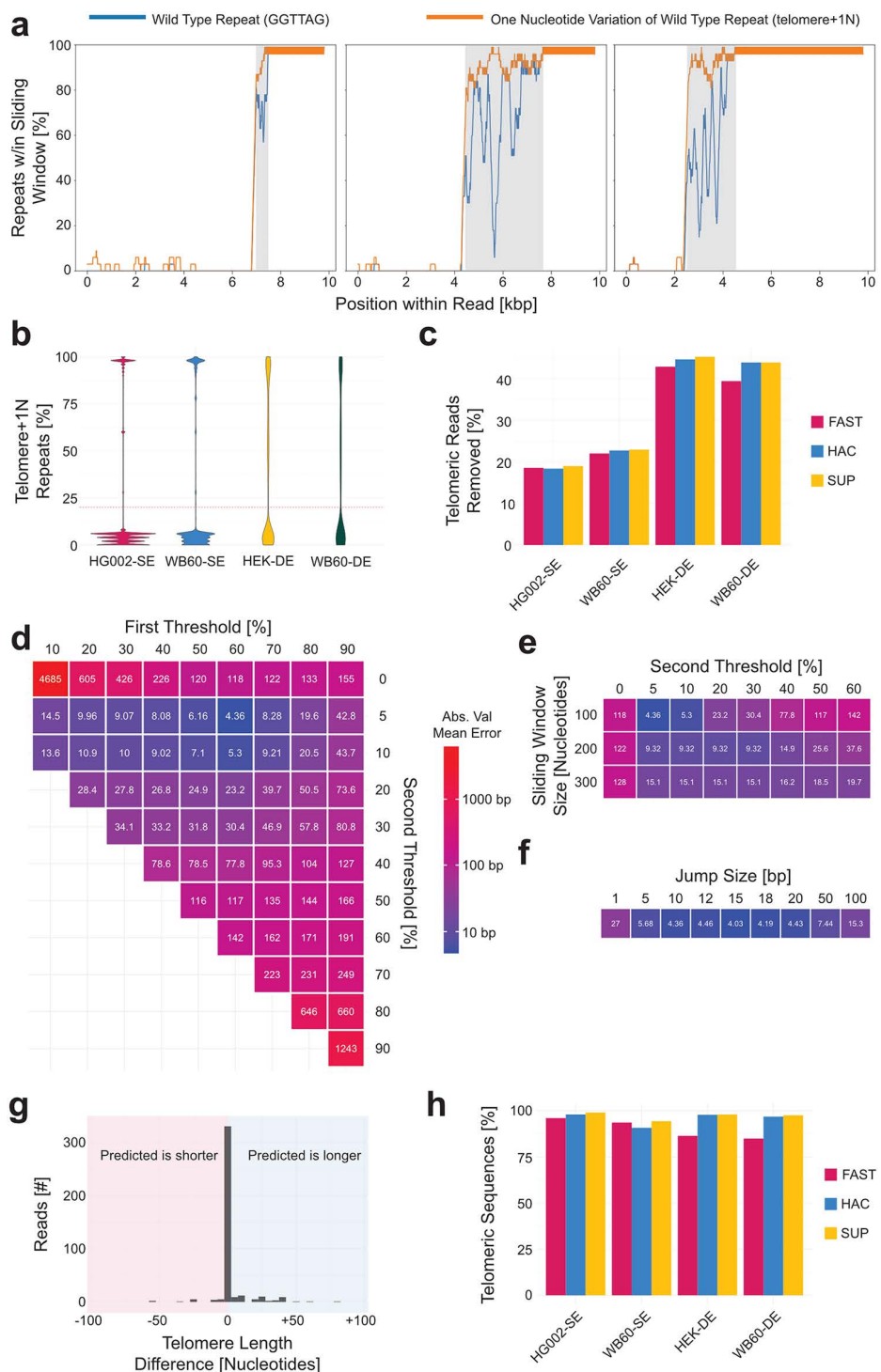

Fig 4. **Identifying the Subtelomere-to-Telomere Boundary. (a)** Three telomeric reads with the percentage of telomere repeats in a 100 bp sliding window (blue lines) and the percentage of telomere+1N repeats in the same window (orange lines). The variant repeat-rich (VRR) region is highlighted in gray. **(b)** The percentage of telomere+1N repeats in the first 300 bp of a read in the subtelomere direction with a 20% threshold indicated by a dotted red line. **(c)** Percentage of telomeric reads removed for containing greater than 20% telomere+1N repeats in the first 300 bp of the sequence. **(d)** Mean absolute value difference in predicted versus manually annotated start of the VRR-region when identifying the start of the VRR-region by the first telomere+1N repeat in a sliding window that exceeds the first threshold while the read does not contain a sliding window that drops below the

second threshold for the remainder of the read. **(e)** Mean absolute value difference in predicted versus manually annotated start of the VRR-region when increasing the sliding window size with a first threshold of 60% and **(f)** increasing the interval size to decrease computational time when a 60%/5% threshold system is used with a sliding window size of 100 bp. **(g)** Differences between manually annotated start sites and computationally identified start site. **(h)** Percentage of telomeric sequences where the VRR-region start site was determined.

a truth set. The accuracy of each algorithm was evaluated by calculating the mean absolute value error between the predicted and manually annotated VRR-region start positions. Lower values indicate higher accuracy.

The first method tested involved identifying the first occurrence of eight consecutive telomere + 1N repeats. This approach yielded a mean absolute error of 149 nucleotides and was ultimately unsuccessful (S3b Fig). A second method used a 100 bp jumping window, applied in 10 bp increments along the read. When the frequency of telomere + 1N repeats in a window surpassed a predefined threshold, the first telomere + 1N repeat within that window was marked as the VRR-region start. To prevent misidentification due to telomere-like islands in the subtelomeric region (S3c Fig), this threshold was required to be maintained throughout the remainder of the read. Using a 60% threshold, this method reduced the mean absolute error to 118 bases (S3d Fig). Further refinement involved a dual-threshold strategy: once the primary 60% frequency threshold was reached, the telomere + 1N repeat content could not fall below a secondary threshold of 5% for the remainder of the telomere. This approach dramatically improved performance, achieving a mean absolute error of 4.36 nucleotides (Fig 4d).

We also tested the effect of window size and step size. Larger window sizes decreased accuracy at low secondary thresholds (Fig 4e), while increasing the jump interval from 10 bp to 15 bp slightly improved precision by ~0.3 nucleotides (Fig 4f). Most VRR-region start sites were predicted with near single-nucleotide accuracy. For those that were inaccurate, the error typically resulted in slight overestimation of telomere length by 10–50 nucleotides (Fig 4g). Using a 100 bp window, a 15 bp increment, and the dual-threshold criteria, the algorithm successfully identified the VRR-region start in over 80% of telomeric reads (Fig 4h).

## Filtering for high confidence telomeric sequences

To ensure that telomeric reads were accurately identified, we examined the proportion of telomere + 1N repeats between the VRR-region start and the capture probe (Fig 5a). On average, this region contained greater than 80% telomere + 1N content. Reads with less than 80% fell into two main categories: (1) sequences in which a telomere start was identified, but the read originated from subtelomeric or interstitial telomeric regions (Fig 5b); or (2) true telomeric reads that included atypical subtelomeric structures, unusually long VRR-regions, and relatively short stretches of invariant telomeric repeats (Fig 5c). To exclude ambiguous or misidentified reads while retaining biologically valid telomeric sequences, we tightened the threshold, removing all reads containing telomeres with less than 60% telomere + 1N content. This removed cases like Fig 5b while retaining complex, valid reads like those shown in Fig 5c. Reads discarded at this step tended to have slightly higher basecalling quality scores (Fig 5d) but were much shorter in both total read length (Fig 5e) and telomere length (Fig 5f) compared to retained reads. These patterns are consistent with a subtelomeric origin, likely representing sequences that failed to extend fully into the VRR- or telomeric region during sequencing. This filtering step excluded fewer than 1% of telomeric reads overall (Fig 5g).

A second validation step was performed to confirm the accurate identification of the VRR-region start by examining the 2 kb of sequence immediately upstream (in the subtelomeric direction) of the predicted boundary. This region was analyzed to ensure it was not composed primarily of telomere + 1N repeats (Fig 5h). Elevated telomere + 1N signal in this region can arise either from genuine subtelomeric structure, consistent across reads (Fig 5i), or from basecalling artifacts (Fig 5j). Reads resembling Fig 5j, which exhibit a sharp transition from fully telomeric sequence to 0% telomere + 1N content, are attributed to sequencing artifacts introduced by dorado v0.7.0. These segments, composed of less than 10%

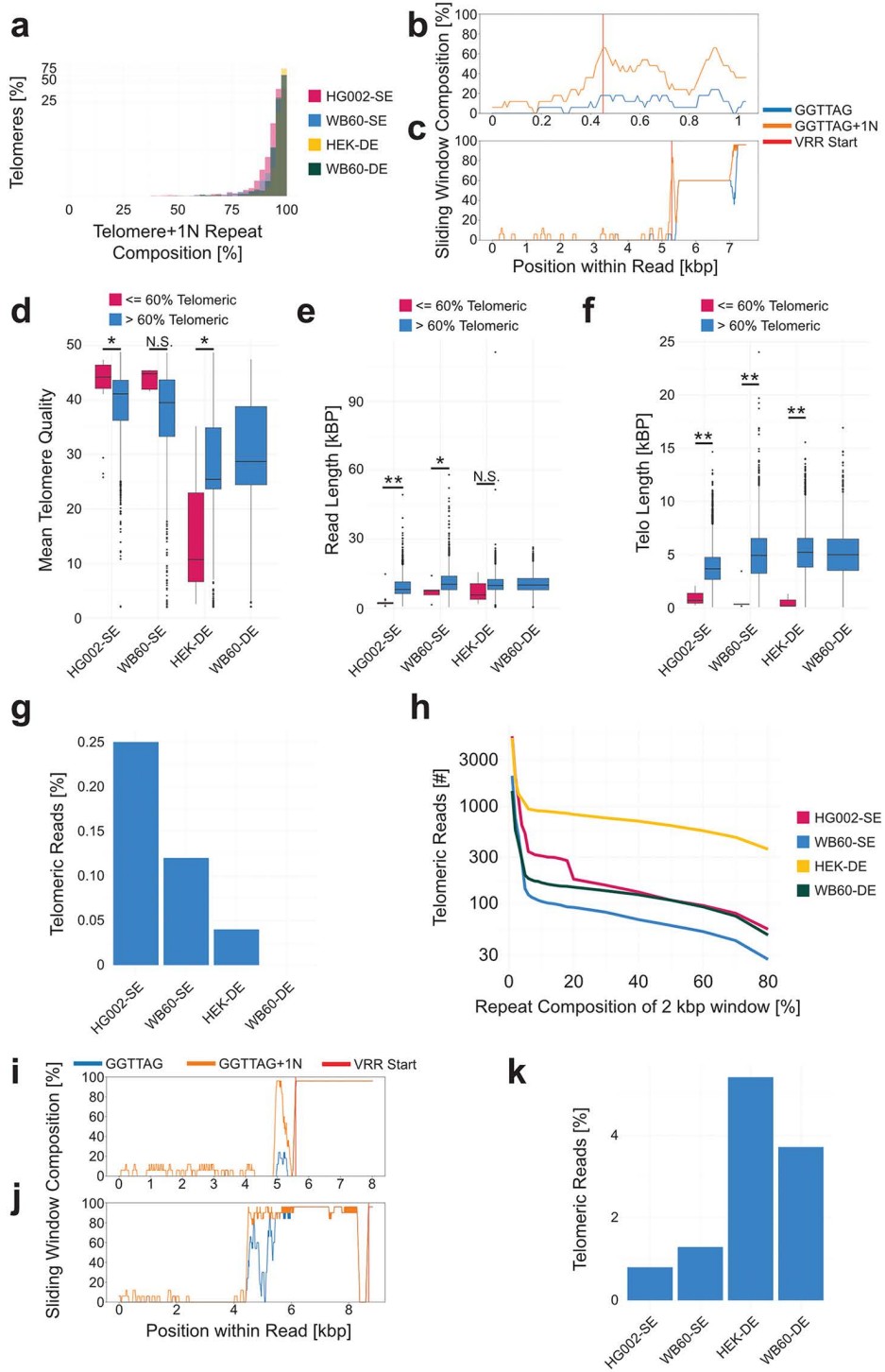

**Fig 5. Filtering of Telomeric Sequences. (a)** Percentage of telomere composed of telomere + 1N repeats. **(b)** An example non-telomeric sequence passing all filtering criteria that is erroneous and **(c)** an example telomeric sequence passing all filtering criteria that is not erroneous and should be retained. **(d)** Mean telomere quality of sequences in which the telomeres are composed of greater than or less than 60% telomere + 1N repeats. **(e)** Read length of filtered telomeric sequences and (f) telomere length of filtered sequences (* = p.val < 0.05, ** = p.val < 0.005). **(g)** Percentage of telomeric sequences removed for being composed of less than 60% telomere + 1N repeats **(h)** Percentage of telomere + 1N repeats in the 2 kb immediately prior to the VRR-region start site in the subtelomeric direction. **(i)** A high telomere + 1N percentage in prior 2 kb read that should not be removed from the

analysis and **(j)** a high percentage read that should be removed from the analysis. **(k)** Percentage of telomeric sequences removed for containing a high proportion of telomere + 1N repeats before the VRR-region start site.

telomere + 1N repeats, typically show reduced quality scores relative to the rest of the telomeric sequence, composed of greater than 85% telomere + 1N repeats, (S4a Fig) and lack consistent VRR-region repeat patterns indicating this is not a chromosome arm-specific or biological phenomena.

To address these cases, the VRR-region detection algorithm was refined while retaining the core thresholds: 60% telomere + 1N content in a 100 bp jumping window, with no drop below 5% for the remainder of the telomere. An additional requirement was introduced: the repeat content must remain below the 5% threshold for at least 15 consecutive windows before concluding that no VRR-region start site is present. This refinement ensures accurate detection in cases like Fig 5i while filtering out artifacts such as Fig 5j. Reads with >10% telomere + 1N content in the 2 kb upstream of the predicted start site were excluded, resulting in the removal of approximately 1%–5% of telomeric sequences (Fig 5k).

The VRR-region is composed primarily of telomeric and telomere + 1N repeats. However, including this region in the calculation of telomere length will result in an increased telomere length when compared to techniques that determine the telomere start solely based on the frequency of invariant telomeric repeats. This difference depends highly on the sequence structure of the VRR-region. There is a mean telomere length difference of $552 \pm 273$ bases when comparing the length of the VRR-region containing telomere to the summation of all wild type telomeric repeats within the same region (S4b Fig).

## Read assignment to specific chromosome arms

Chromosome arm-specific telomere length differences have been reported in the budding yeast *Saccharomyces cerevisiae* [36,37] and underlying regulatory mechanisms have recently been identified. These findings have sparked considerable interest in applying chromosome arm-level telomere analysis in other model organisms and humans. Accurate assignment of each telomeric read to a unique chromosome arm is therefore an important goal and only high quality, SUP basecalled data should be used to identify chromosome arm specificity.

Telomeric sequences were aligned to a high-quality, HG002-specific subtelomere reference generated by extracting the terminal region of each chromosome arm from the telomere to the first EcoRV restriction enzyme digest site from the telomere-to-telomere (T2T) HG002 v1.0 reference. Only the terminal region of the reference subtelomere was used to prioritize alignment on regions adjacent to telomeres. Alignment was performed using minimap2 v2.26-r1175 and default parameters with the mapping option "-ax map-ont" specified. Alignments were then filtered for a minimum mapping quality (MAPQ) of 0–10, 20, 40, and 60 prior to determining the number of uniquely mapping reads (reads that have only one alignment with a MAPQ equal to or greater than the specified quality score), unmapped reads (reads with no alignments), and multimapping reads (reads with multiple alignments with a MAPQ equal to or greater than the specified quality score). Aligning telomeric reads from HG002 cells to the generated reference enables assignment of over 90% of telomere-containing reads to individual chromosome arms, even when a strict mapping quality of 60 is used (Fig 6a, HG002-SE, solid red line). However, aligning the same dataset to other subtelomeric references or general-purpose T2T assemblies (e.g., STONG [38] or CHM13 [39]) yields substantially lower performance: only ~60% of reads map uniquely, many are multimapping with low quality scores, and the proportion of unmapped reads increases as the minimum mapping quality threshold is raised. This issue is further illustrated here when aligning telomeric reads from non-HG002 samples—such as the clinical sample WB60 (SE and DE libraries)—to the HG002, CHM13, or STONG references. In these cases, only ~50% of telomeric reads map uniquely with a MAPQ ≥ 10, indicative of poor alignment specificity (Fig 6a, WB60-SE). Data derived from the HEK293T cell line was not included for analysis due to its hypotriploid status.

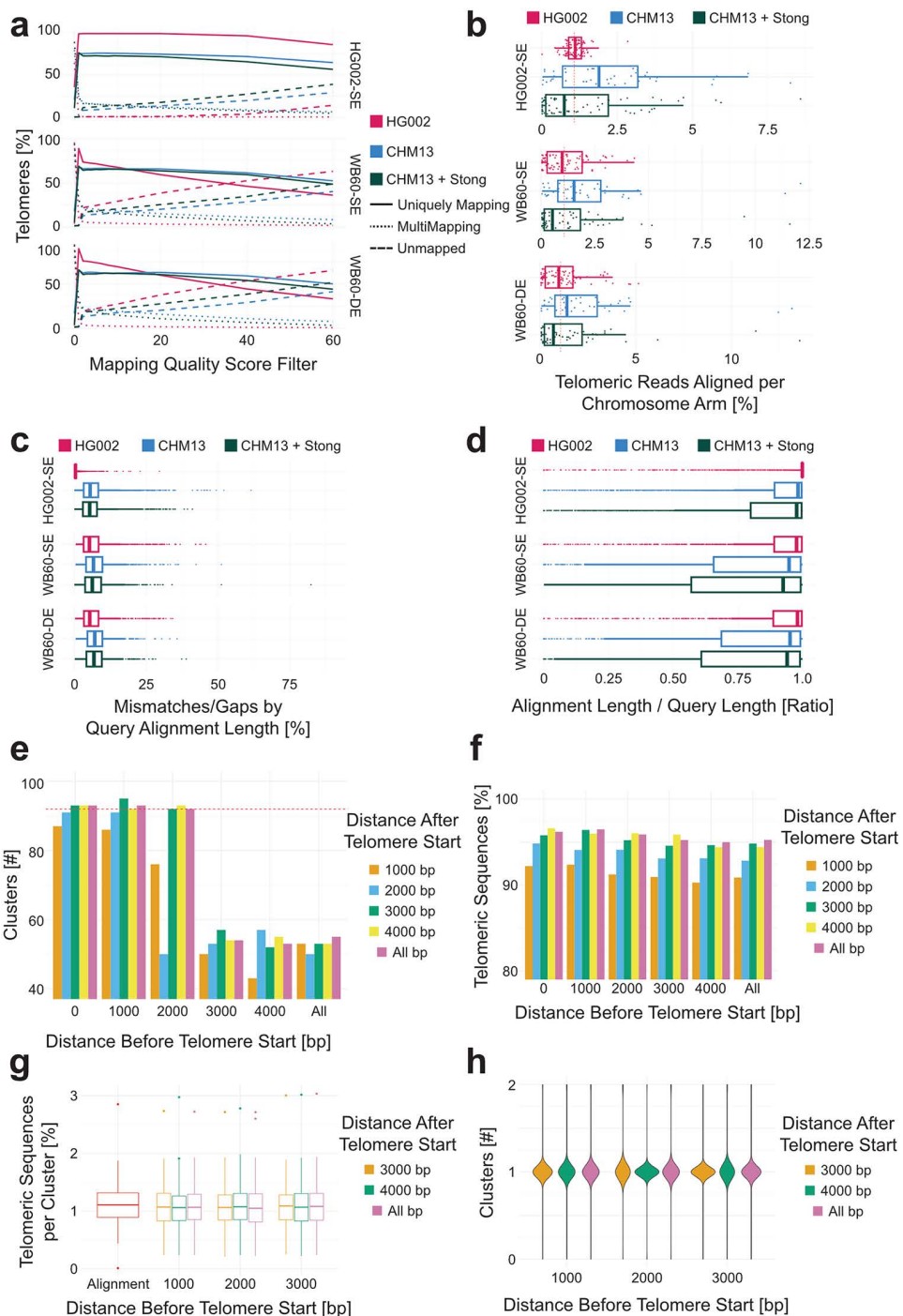

**Fig 6. Chromosome arm-specific telomere length. (a)** Number of uniquely mapping reads, multimapping reads, and unmapped reads of three different telomeric sequence datasets aligned to three different subtelomeric reference genomes. **(b)** The chromosome arm distribution of telomeric sequences when aligning to different reference genomes with the expected 1.08% of sequences mapping to each chromosome arm annotated with a dotted red line. **(c)** The percentage of the alignment length that is composed of gaps or mismatches when aligning the three datasets to three different subtelomeric references and **(d)** the length of the alignment as a ratio to the length of the query sequence. **(e)** Number of clusters composed of greater than 0.02% of the input telomeric sequences when trimming the telomeric sequences to consistent lengths. The dotted red line represents the expected number of clusters for a diploid human cell (92). **(f)** The percentage of input telomeric sequences contained within one of the clusters documented in panel e. **(g)** The distribution of telomeric sequences across the clusters directly compared to the distribution of the same telomeric sequences aligned to

the HG002 subtelomeric reference genome (red). **(h)** The number of clusters that align to each chromosome arm in the HG002 subtelomeric reference in relations to the distance before and after the telomere start site. One indicates that all reads aligning back to the same chromosome arm are found within a single cluster.

Furthermore, since neither enrichment technique has an intrinsic bias against specific chromosome arms, an even distribution of telomeric reads across all 92 chromosome arms should be expected. This was indeed observed when aligning HG002-derived telomeric reads to the HG002-specific reference: approximately 1.08% (dotted red line) of telomeric sequences uniquely aligned to each arm, with no arm exceeding 3% of the total sequences when filtering for reads with an alignment MAPQ ≥ 10 (Fig 6b).

In contrast, alignments of the same HG002 dataset to other reference genomes (e.g., CHM13 or CHM13 + STONG) introduced significant bias with greater than 7.5% of the telomeric reads sequenced aligning uniquely back to a single chromosome arm (Fig 6b). The effect was also seen in clinical samples aligned to the same references, where individual chromosome arms contained greater than 10% of the telomeric reads aligning (Fig 6b). These distribution disparities were accompanied by higher mismatch and gap counts when aligning clinical datasets to non-matching references, as compared to HG002 aligned to its own reference (Fig 6c). The poor alignment quality can additionally be seen when comparing the subtelomeric alignment length against the subtelomeric length of the query sequence (Fig 6d). While this ratio is near 1 for HG002 telomeres aligned to the HG002 reference, this decreases substantially for other reference genomes or clinical samples. Interestingly, while few differences exist in a chromosome arm specific manner in the alignment of HG002 sequences (S5a Fig), certain chromosome arms in clinical samples (Figs S5b and 5c) exhibit more complete subtelomeric alignments than others. It remains to be seen if this is simply an artifact of alignment or a global conservation in certain subtelomeric structures. These results clearly demonstrate that aligning telomeric sequences to a reference genome derived from a different individual or cell line does not provide reliable chromosome arm-specific telomere measurements.

To test whether de novo clustering techniques can provide telomere allele-specific length information, telomeric sequences derived from HG002 sequencing data were de novo clustered and then compared to the alignment of the same sequences to the HG002 subtelomeric reference. Parameters optimized with HG002 were then applied to other datasets for validation. Telogator2 (commit #d4e50d1) [31,40,41], a pipeline designed to derive telomere allele-specific lengths from long reads, can be used to cluster telomeric reads without alignment to a reference using pairwise alignment and hierarchical clustering. Perfect clustering would result in the formation of 92 clusters with 100% of the input telomeric sequences divided evenly across the clusters (~1.08% of the sequences per cluster). A cluster is retained for further analysis only when containing greater than 0.02% of the input telomeric sequences. This removes very small artificial clusters created by Telogator2 composed of as few as one or two telomeric reads. When full length telomeric sequences were used as input for Telogator2 (-tt 0.1 --collapse-hom 500 -r ont -p 10 --filt-tel 0 --filt-nontel 10000 --filt-sub 0 –debug-noanchor) with parameters designed to turn off read filtering as TARPON has already identified if the reads are telomeric or not, only 55 clusters were generated (Fig 6e) with ~40% of the telomeres clustering together in a single allele (S6a Fig, right).

To focus clustering primarily on the VRR-region, telomeric sequences were trimmed to the sequence found immediately before or after the telomere start identified by TARPON. When reads were trimmed to contain 3 or 4 kb of subtelomeric sequence (distance before telomere start) and clustered, similar trends to full-length reads are observed with fewer clusters than expected, regardless of the length of telomeric sequence (distance after telomere start) (Fig 6e). However, including 0 kb, 1 kb, or 2 kb of subtelomeric sequence (before the telomere start), resulted in a notable increase in the number of identified clusters (Fig 6e). Increasing the length of telomeric sequence included within the reads (distance after start) increases the percentage of telomeric sequences included in the clusters which contain greater than 95% of the

telomeres sequenced when using 1 kb before the telomere start and 3 or 4 kb after the telomere start (Fig 6f). The length of subtelomeric sequence used directly influences the number of telomeric clusters created (Fig 6e), while the length of telomeric sequence used effects the number of telomeric sequences included in the clustering results after exclusion of clusters containing less than 0.02% of the telomeric sequences.

When telomeres are trimmed to contain 1 kb, 2 kb, or 3 kb of subtelomeric sequence (distance before the telomere start) and either 3 or 4 kb telomeric sequence (after the telomere start), the telomeric sequences are distributed equally across the telomere alleles, directly comparable to the chromosome arm distribution seen by aligning the same reads to the HG002 reference (Fig 6g). Furthermore, the read IDs belonging to each chromosome arm by alignment are uniquely found in a single cluster (Fig 6h) demonstrating the accuracy of the clustering methodology. Additionally, visualization of the clusters by plotting individual telomeric sequences and comparing the VRR-region structure further supports the accuracy of the clustering methodology (S7a Fig).

When the parameters established for HG002 were tested on clinical samples the results varied with the number of telomeric sequences used. For WB60-DE and WB60-SE only 2,034 and 3,170 telomeric reads were available, respectively. Here we see a slightly higher cluster count than expected, 94 as opposed to 92 (S6b Fig) and a slightly decreased percentage of telomeric sequences included in clusters (~90%) (S6c Fig) indicating that the clustering parameters may need to be fine-tuned in a sample-specific context. Nevertheless, the results from the clustering approach were superior compared to the alignment when using a non-sample-specific reference. It is recommended that at least 1 kb of subtelomeric sequence (before the telomere start) is included in the analysis to avoid formation of large clusters due to high sequence similarity within the VRR-region (S6d Fig).

The computational time required for Telogator2 is substantially larger than performing a simple alignment and performance of the software varies with parallelization; best results were identified when using a minimum of 10 threads. The telomere allele-specific clusters are not referred to as chromosome arm-specific clusters as it is currently impossible to assign these clusters to a specific chromosome arm without a high-quality subtelomeric reference genome, not available for non-HG002 samples in this study.

## Extended features of TARPON

Beyond its default configuration, TARPON includes a range of optional parameters for more specialized analysis of telomeric sequences and enrichment protocols. For duplex-enriched datasets, the --strand_comparison flag provides strand-specific enrichment metrics, including relative abundance, filtering outcomes, and telomere length distributions for C- and G-strand reads. The --detailed_stats option outputs additional read-level statistics and visualizations, such as repeat composition and telomere length-to-quality score comparisons, in text, graphical, and HTML formats.

For users exploring telomeric enrichment strategies, the --restriction_digest flag accepts a comma-separated list of restriction enzyme recognition sites and returns the number of telomeric sequences affected by each restriction site. In samples containing mutant telomeric repeats such as in the case of a telomerase RNA template mutation [42], the --mutant flag enables these sequences to be included in all filtering and boundary-identification steps, while also returning statistics related to mutant versus wild type telomerase processivity.

To reduce computational overhead, TARPON also supports a hybrid basecalling strategy. When --pod5_directory is specified, telomeric reads are first isolated from fast basecalled data and then re-basecalled using SUP models. Input file formats do not need to be modified in advance: FASTQ and BAM files are both accepted, and TARPON will convert files to UBAM format internally as needed. Integration with Nextflow and the use of Docker containers or conda environments helps prevent versioning issues and dependency conflicts during execution.

## Comparison with other telomere analysis software

While several software tools are available for analyzing telomere content from Illumina whole-genome sequencing (WGS) data, only TeloBP [29], Telometer v1.1 [27], and wf-teloseq v0.1.0 have been designed specifically for nanopore-based

telomere sequencing with the latter being released by ONT while this manuscript was in preparation. TeloBP and Telometer both require experience with command-line interfaces, manual installation of dependencies, and troubleshooting version conflicts. Neither TeloBP nor Telometer provides a complete analysis pipeline: users must first isolate and demultiplex telomeric reads, remove low-quality or chimeric sequences, and ensure the presence of a capture probe prior to estimating telomere length or identifying the subtelomere-to-telomere boundary.

In Telometer, reads must be aligned to a subtelomeric reference genome, and any read that fails to map within the first or last 30 kb of the reference is discarded. TeloBP requires input in FASTQ or compressed FASTQ format and filters out chimeric reads before proceeding. Both tools perform limited quality filtering: Telometer excludes reads with an average quality score below 9 and TeloBP relies on external preprocessing.

In contrast, TARPON and wf-teloseq are accessible both via command-line and through a graphical user interface (GUI) (ONT's EPI2ME) for users less familiar with scripting. Nextflow's integration with Docker containers eliminates dependency issues entirely. TARPON accepts fast or SUP basecalled output directly from the sequencer, requiring no data preprocessing. Reads must be demultiplexed prior to executing wf-teloseq while TARPON will internally handle all sample demultiplexing. Filtering steps prior to subtelomere-to-telomere boundary identification also differ between TARPON and wf-teloseq. While both identify a capture probe at the end of the telomere, wf-teloseq does so indirectly by requiring all input reads to be demultiplexed. TARPON ensures the entire telomere has been sequenced and the read is not chimeric prior to telomere boundary identification, while wf-teloseq filters incompletely sequenced telomeres after boundary identification. Additionally, wf-teloseq requires a minimum read length of 120 bp and at least 100 telomeric repeats: this parameter is not customizable and effectively requires a telomere to be composed of at least 600 canonical telomeric nucleotides, potentially eliminating a subgroup of telomeres with high biological relevance.

While TeloBP, Telometer, and TARPON all use a jumping window approach to identify the subtelomere-to-telomere boundary, only TARPON allows users to adjust relevant parameters without modifying source code. wf-teloseq uses a sliding window approach through the linear convolution of two arrays of different sizes but does not allow for any user flexibility. Customization in TeloBP and wf-teloseq requires editing the Python files within the cloned GitHub repository. For Telometer, users must first locate the pip package installation and modify Python scripts within those directories. By contrast, TARPON supports parameter customization either through standard Nextflow command-line syntax or via the EPI2ME GUI prior to workflow execution, providing more accessible and flexible control over the boundary detection process.

The details of how each tool defines and filters telomeric regions also vary. Telometer uses a 120 bp sliding window that advances in 12 bp increments to identify telomeric regions. If GGTTAG repeats make up more than 10% of a window, the start of a telomeric region is defined. If a subsequent window falls below the 10% threshold, a gap is introduced. If this gap exceeds 100 nucleotides and the frequency of telomere + 1N repeats within the gap is also less than 10%, the telomeric region is terminated. A new telomeric region may then be defined further along the read. After read processing, Telometer applies several filtering steps. If no telomeric regions were identified, or if no window within a telomeric region contains more than 75% GGTTAG, the read is excluded from further analysis. If the average quality score of the gap between two telomeric regions is less than or equal to 9, any telomeric region following that gap is discarded. Telomeric regions are also excluded if they do not begin or end within 100 bp of a read boundary. Telomere length is then calculated from the first telomeric region that passes all filtering criteria. Finally, if the estimated telomere length plus 50 bases exceeds the total read length, the read is omitted from further analysis.

TeloBP uses a similar but more complex approach to identify the subtelomere-to-telomere boundary. When executed with default parameters, a 100 bp jumping window with 6 bp intervals scans the read in the telomere-to-subtelomere direction for the presence of "GGG" motifs. For each window, the deviation from the expected composition (50%) is calculated as: (observed − expected)/expected × 100. This value is then plotted, and the area under the curve (AUC) is computed using a series of 83 sliding windows. If the AUC in a given window is less than –50 (suggesting the region is not telomeric) and the next window contains fewer or equal telomeric repeats, this defines a new threshold from which further analysis

begins. From this threshold, the differences in AUC values between adjacent sliding windows are calculated. If the absolute difference between two AUC values is less than 0.2, or if the current difference is less than 0.2 and the next difference exceeds 0.2, the boundary between the telomere and subtelomere is defined. Telomere length is then estimated as the distance from this boundary to the end of the read.

In contrast to the use of sequence context by Telometer and TeloBP, wf-teloseq first converts the telomeric sequence into a binary array: 1 for telomeric sequence or telomeric variants and 0 for non-telomeric sequence. wf-teloseq does not refer to variants as telomere + 1N repeats, but as basecalling variants, i.e., CACCCT, ACCCCT, CCCAAA, CCCCGA, etc. and greatly reduces the length of the VRR-region. After binary sequence conversion, the resultant array is smoothed by scipy.ndimage.median_filter. If there is a high enough density of wild type telomeric repeats within the variant repeat-rich region, the smoothing may allow for the capture of the VRR-region. However, this smoothing occurs in a sequence-specific manner and would affect each chromosome arm differently creating a systematic bias for chromosome arm-specific telomere length analysis. The smoothed binary array is then compared to a mock telomere boundary (an array of 61) composed of 30 nucleotides of telomeric sequences (1 after binary conversion), a 0 value, and then 30 nucleotides of -1 value. These two arrays are compared using np.convolve which returns the arrays discrete, linear convolution. The telomeric boundary is identified as the last occurrence of the minimum value found within the newly calculated convolution. As wf-teloseq is designed to operate on C-strand telomeric sequences only, the final occurrence within a read would be centromere proximal.

Since each software utilizes a slightly different approach than TARPON to identify the subtelomere-to-telomere boundary, the 400 manually curated telomeric sequences were used to calculate the mean absolute value error of wf-teloseq as 157 nucleotides, of Telometer as 161 nucleotides, and of TeloBP as 29.8 nucleotides, compared to the 4.03 nucleotide mean absolute value error of TARPON (S7b Fig). Additionally, 6 reads within Telometer and 10 reads within wf-teloseq had an absolute value error greater than 500 nucleotides with a maximum error of 10,140 nucleotides in wf-teloseq driven primarily by the identification of telomere islands within the subtelomeric sequences (S7c Fig). While these telomere islands may contribute to chromosome stability and exhibit shelterin binding, the intervening non-telomeric sequence most likely does not and would result in a non-biological chromosome-arm-specific telomere length bias.

After boundary detection, TARPON and wf-teloseq apply additional filtering steps not found in TeloBP or Telometer. In TARPON, the telomeric region must contain more than 60% telomere + 1N repeats, and the 2 kb region upstream of the boundary must contain less than 10% telomere + 1N. These filters eliminate reads with internal sequencing artifacts that arise through erroneous basecalling and exhibit reduced quality scores, such as the example in Fig 5j. In this case, TeloBP and Telometer assign telomere lengths of 18 bp and 342 bp, respectively. In reality, the read contains over 3 kb of telomeric sequence, but since it is unclear if the erroneous stretch of basecalls is similar in length to the bona fide repeat sequence, the read is excluded from analysis in TARPON. wf-teloseq first ensures the telomere boundary identified is greater than 61 nucleotides away from the start of the read and 30 nucleotides away from the end of the read. While the README of wf-teloseq states the distance as 60 nucleotides from the end of the read, within the source code this parameter is divided by 2 during the condition operation. The first 30% of the identified telomere (C strand sequences start at the distal end of the telomere) must be composed of greater than 80% telomeric repeats. The subtelomeric portion of the telomeric read is then confirmed to be composed of less than 25% CCC and a median sequence quality greater than 9. Lastly, wf-teloseq filters for known telomere basecalling artifacts and if it finds 5 of such artifacts within 500 bp of each other the read is discarded from the analysis.

TARPON calculates telomere length up to the start of the capture probe and wf-teloseq calculates telomere length as the distance between the capture probe, trimmed off during demultiplexing, and the identified telomere boundary. In contrast, if reads are not preprocessed before using TeloBP, telomere length estimates will be inflated due to the inclusion of non-telomeric sequence contributed by the capture probe and/ or ONT sequencing adapters.

Telometer, TeloBP, wf-teloseq, and TARPON all return tabular output files (CSV or TSV) listing read IDs and corresponding telomere lengths. TARPON includes additional metadata such as the coordinates of the telomeric region within each read, strand specificity, and read-level quality metrics. It also generates a suite of visual outputs summarizing pipeline execution, filtering steps, and telomere length distributions. A precompiled HTML report—automatically launched in the EPI2ME GUI upon completion—provides an accessible, sample-by-sample overview. For multiplexed datasets, side-by-side comparisons are included by default. While TARPON and wf-teloseq are both complete analysis pipelines (excluding the lack of demultiplexing in wf-teloseq), they differ in key functionalities: wf-teloseq is only applicable to C-strand telomeric sequences, offers no user flexibility making it difficult to adapt to specific use cases such as ALT-positive cell lines which may contain a higher proportion of variant telomeric repeats. Additionally, the discrepancies that exist between the source code of wf-teloseq and the README make it difficult for non-computational users to identify how their telomeric sequences are being processed to ensure no bias is being introduced. An overview of the functionalities of each pipeline is available in S8 Fig.

## Availability and future directions

TARPON is a flexible and modular pipeline designed to analyze telomeric sequences from ONT long-read sequencing data. It performs a full analysis workflow from fast or SUP basecalled data, including telomeric read isolation, capture probe and barcode identification, and subtelomere-to-telomere boundary detection. The output includes quality metrics and telomere length statistics in graphical and tabular form.

While default parameters are optimized for telomerase-positive human samples, all settings can be customized to accommodate different organisms, enrichment strategies, or specific experimental goals. This flexibility allows TARPON to support a wide range of research contexts, including species with noncanonical telomere repeats, mutant telomerase variants, and different enrichment chemistries. Future releases of TARPON will support additional features such as strand bias analysis, telomeric variant detection, and methylation incorporation.

Here, the authors presented two simplex enrichment and two duplex enrichment sequencing experiments. While a large fraction of G-strand telomeric reads in the duplex enrichment methodology do not contain a capture probe and are therefore removed from the analysis, no notable differences in final telomere read count were apparent and read count differences are more likely a consequence of flow cell health than enrichment protocol differences. However, differences between DNA extraction methods greatly impacts the number of telomeric reads sequenced with SE_HG002 and DE_HEK containing 8,286 and 7,520 telomeric reads, respectively, while the robotically extracted, lower molecular weight DNA of SE_WB60 and DE_WB60 results in 3,171 and 2,034 telomeric sequences, respectively. Further analysis to understand the influence of DNA extraction quality on telomere enrichment will be necessary to ensure maximal protocol efficacy.

## Chromosome arm-specific analysis

Despite their workflow differences, all previously published analysis tools rely on alignment to a reference genome for chromosome arm assignment. This method works well for HG002 data aligned to an HG002-specific reference but yields poor accuracy for other cell lines or clinical datasets, where alignment to non-matching references leads to increased mapping bias and reduced reliability in chromosome arm-specific telomere length estimation. TARPON is the first pipeline to provide a solution to this dilemma.

While pre-existing tools to cluster telomeric sequences such as Telogator2 exist for measuring allele-specific telomere length, performance was underwhelming when using full-length telomeric sequences in early 2025. However, when focusing on the variant regions that differ between telomere alleles (the variant repeat-rich region) between 90 and 95 clusters of equal size were obtained even with low input read counts. It is important to note that Telogator2 inherently maps clusters back to a subtelomeric-specific reference; however, when executed within TARPON this is not allowed and Telogator2 is terminated after cluster formation. Telogator2 utilizes pairwise alignment and hierarchical clustering resulting

in long run times. In the future, other programs that can accurately assign de novo telomeric reads in a cluster-specific manner should be evaluated. Additionally, while Telogator2 (commit #d4e50d1) served as a solid foundation for de novo clustering, newer versions and other software that may decrease computation requirements or increase accuracy should be explored.

### Multiplexing and cost reduction

At the time of manuscript preparation, the cost to sequence a telomere enriched sample was €95 for nanopore specific library preparation reagents (SQK-LSK114), €34 for third party reagents (NEBNext Companion Module v2, E7672S), and €570 for a R10.4.1 MinION flow cell assuming purchase of a pack of 12 flow cells, plus the cost of sample generation/collection/enrichment. This results in a minimum total cost per sample of 700€. To reduce this considerable cost per sample, it is crucial that reliable multiplexing methods are developed without decreasing the number of telomeric reads per sample. The multiplexing protocols described previously [27–29] allow for sample pooling after duplex or splint ligation prior to library preparation. Multiplexing introduces additional computational challenges, which TARPON already addresses. If a sample file is provided that contains unique barcode sequences (often found within the duplex sequence as in Karimian and colleagues [29]) together with a capture probe, the capture probe sequence is first used to identify the terminal end of the telomere. The following 100 bp are then used to demultiplex the samples based on the barcodes found within the sample file. If no capture probe sequence is provided, but a sample file is, the barcodes found within that sample file will be used to both demultiplex the data and determine the end of the telomere. If only a capture probe is provided without a sample file, TARPON assumes the dataset was not multiplexed.

### Applications of nanopore telomere sequencing

Assessing telomere length distributions at nucleotide resolution opens new avenues for studying telomere dynamics in aging and senescent cell populations. Nanopore sequencing enables a clearer definition of the subtelomere-to-telomere boundary and provides the opportunity to investigate the functional relevance of the variant repeat-rich (VRR) region. It also allows for high-resolution studies of cancer cells exhibiting recombination-based telomere maintenance or elevated telomere + 1N repeat content, as well as the detailed characterization of the effects of telomerase template mutations on telomere dynamics.

The utility of nanopore-based telomere sequencing extends well beyond traditional research. FlowFISH, the current clinical gold standard for telomere length diagnostics, provides only a median telomere length per sample and is unavailable in many clinical settings. Nanopore sequencing, by contrast, offers detailed telomere length distributions and can be performed in any laboratory equipped with a MinION device, significantly reducing turnaround time for clinical assessments.

Importantly, telomere analysis with TARPON is not limited to human samples. TARPON can be used to study telomere length dynamics in any organism with non-heterogeneous telomeric repeats, including many invertebrates and plant; however, these uses remain untested. However, we want to caution that parameters for boundary detection may require optimization on a species-specific basis. For non-vertebrate taxa, where telomere repeats can be highly heterogeneous, such as in the fission yeast *Schizosaccharomyces pombe* [43], TARPON is not currently recommended. In such cases, the soon to be released pombeTARPON, designed to account for sequence heterogeneity, will be more appropriate. Additionally, while TARPON is designed for easy clinical implementation Nanopore sequencing nor TARPON have been clinically validated and should at this time not be the sole clinical diagnostic methodology.

### Conclusion

Nanopore sequencing is a rapidly evolving technology that offers a unique opportunity to explore telomere-related questions previously inaccessible with Sanger, short-read Illumina, or PacBio sequencing platforms. While several tools exist for analyzing telomeric sequences from short-read whole-genome sequencing data, few address the distinct challenges

posed by nanopore reads. Among those that do, most require bioinformatics expertise and substantial preprocessing, and are limited to command-line interfaces.

TARPON is the first fully automated and GUI-accessible telomere analysis pipeline tailored to nanopore sequencing. It supports both splint- and duplex-enriched telomeric libraries and is designed for ease of use with experimentally validated defaults and seamless integration into the EPI2ME platform. No command-line experience or manual data manipulation is required for standard operation. At the same time, TARPON offers advanced users full flexibility to adjust parameters for specialized research questions, including non-human samples and atypical telomeric features.

By generating accessible tabular and graphical outputs, including a complete HTML report, TARPON empowers researchers and clinicians alike to analyze telomere length with precision and transparency. The pipeline is publicly available at https://github.com/baumannlab/TARPON.

## Supporting information

**S1 Fig. The percentage of telomeric sequences from four sequencing runs that contain a capture probe separated by strand.** HG002-SE and WB60-SE should not contain G-strand telomeric sequences as the enrichment protocol used should result in only C-strand telomeric sequencing.
(TIFF)

**S2 Fig. (a)** Three examples of reads that end in a capture probe but lack a region of invariant telomeric repeats, instead terminating within the variant repeat-rich regions. Blue lines represent the frequency of GGTTAG repeats within a 100 bp sliding window, orange lines represent the frequency of all telomere + 1N repeats within a 100 bp sliding window, and red lines represent the VRR-region start site.
(TIFF)

**S3 Fig. (a)** Percentage of telomeric sequences that contain less than 20% telomere + 1N repeats in the first 300 bp of the read opposite of the capture probe separated by strand. **(b)** Absolute value mean error of the subtelomere to telomere boundary of 400 manually annotated reads defined by a stretch of consecutive telomeric repeats of a given length. **(c)** An example telomeric sequence that contains a telomere-like island within the subtelomere represented by an increased frequency of telomere + 1N repeats approximately 3.8 kb into the sequence where the blue line represents the frequency of wild type telomeric repeats in a 100 bp sliding window and the orange line represents the frequency of telomere + 1N repeats in the same window. **(d)** Absolute value mean error of the subtelomere to telomere boundary of 400 manually annotated reads defined by the first sliding window to be composed of greater than a given percentage of telomere + 1N repeats.
(TIFF)

**S4 Fig. (a)** Disitrbution of the average quality score in 100 bp segments of telomeric sequences that are composed of greater than 85% telomere + 1N repeats (real telomeric sequences) and sequences that are composed of less than 10% telomere + 1N repeats after the start of the telomere is identified. Sequences composed of less than 10% telomere + 1N repeats are resultant of basecalling artifacts as seen in Fig 5j. These reads are ultimately removed from analysis by TARPON. **(b)** The difference between calculating telomere length from the subtelomere-to-telomere boundary to the end of the read compared to the number of nucleotides consisting of wild type telomeric repeats within the same region for all HG002-SE telomeric reads passing all filtering criteria.
(TIFF)

**S5 Fig. The chromosome arm-specific ratio of alignment length to query length of the subtelomeric portion of telomere-containing sequences that are uniquely aligned via Minimap2 to the HG002 subtelomeric reference**

when aligning **(a)** HG002 telomeric sequences, **(b)** WB-60 SE telomeric sequences, and **(c)** WB-60 DE telomeric sequences.
(TIFF)

**S6 Fig. (a)** Percentage of telomeres aligning back to each chromosome arm or present in each cluster when full-length telomeric sequences are passed to Telogator2. **(b)** The number of clusters when non-HG002 samples are clustered using Telogator2 and **(c)** the percentage of telomeric reads composing said clusters. **(d)** The distribution of telomeres across all clusters for non-HG002 samples.
(TIFF)

**S7 Fig. (a)** Five example reads from three randomly chosen clusters showing the variant repeat-rich region pattern is identical in a cluster specific manner. Blue lines represent the frequency of GGTTAG repeats and orange lines represent the frequency of telomere + 1N repeats in a 100 bp sliding window. **(b)** A comparison between the four described telomere analysis software in the accuracy of telomere length prediction compared to the manual annotation of 400 telomeric sequences. **(c)** The behavior of wf-teloseq in the presence of a subtelomeric island that results in the edge of the island being identified as the telomere start site.
(TIFF)

**S8 Fig. A comparison of TARPON, wf-teloseq, Telometer, and TeloBP.**
(TIFF)

**S1 Table. Oligos used in the enrichment of telomeric sequences.**
(XLSX)

**S1 Methods. Supplemental Methods. (a)** A description of the samples presented in this study and where appropriate the DNA extraction techniques used. **(b)** The splint-based telomere enrichment strategy employed in this study for samples HG002-SE and WB60-SE. **(c)** The duplex-based telomere enrichment strategy employed in this study for samples HEK293-DE and WB60-DE. **(d)** Relevant parameters to the basecalling of raw Nanopore sequencing data and the execution of TARPON.
(DOCX)

## Acknowledgments

The authors would like to thank Dr. Lars Erichsen for culturing HEK293T cells, the lab of Prof. Susann Schweiger for genomic DNA of a 60-year-old individual, Robert Vettel, at the Institute for Quantitative and Computational Biosciences (IQCB) for systems administration and members of the Baumann Laboratory for insightful discussions. We thank the Institute for Quantitative and Computational Biosciences, the Nucleic Acid Core Facility at JGU, and the Computational Systems Genetics Group at UMC for computing resources.

## Author contributions

**Conceptualization:** Nathaniel Deimler, Zoë Gill.

**Data curation:** Nathaniel Deimler, David V. Ho.

**Formal analysis:** Nathaniel Deimler.

**Funding acquisition:** Peter Baumann.

**Investigation:** Nathaniel Deimler.

**Methodology:** Nathaniel Deimler, David V. Ho, Zoë Gill.

**Project administration:** Norbert Paul, Peter Baumann.

**Resources:** Peter Baumann.

**Software:** Nathaniel Deimler.

**Supervision:** Norbert Paul, Peter Baumann.

**Validation:** Nathaniel Deimler.

**Visualization:** Nathaniel Deimler.

**Writing – original draft:** Nathaniel Deimler, Peter Baumann.

**Writing – review & editing:** Nathaniel Deimler, David V. Ho, Norbert Paul, Zoë Gill, Peter Baumann.

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
