## [Decision Letter · Decision Letter 0]

22 Oct 2025

PCOMPBIOL-D-25-01744

TARPON - a Telomere Analysis and Research Pipeline Optimized for Nanopore

PLOS Computational Biology

Dear Dr. Deimler,

Thank you for submitting your manuscript to PLOS Computational Biology. After careful consideration, we feel that it has merit but does not fully meet PLOS Computational Biology's publication criteria as it currently stands. Therefore, we invite you to submit a revised version of the manuscript that addresses the points raised during the review process.

Please submit your revised manuscript within 60 days Dec 22 2025 11:59PM. If you will need more time than this to complete your revisions, please reply to this message or contact the journal office at ploscompbiol@plos.org. Please include the following items when submitting your revised manuscript:

We look forward to receiving your revised manuscript.

Kind regards,

Adam Ewing

Academic Editor

PLOS Computational Biology

Ferhat Ay

Section Editor

PLOS Computational Biology

**Journal Requirements:**

2) Your manuscript is missing the following sections: Design and Implementation. Please ensure that your article adheres to the standard Software article layout and order of Abstract, Introduction, Design and Implementation, Results, and Availability and Future Directions. For details on what each section should contain, see our Software article guidelines:

https://journals.plos.org/ploscompbiol/s/submission-guidelines#loc-software-submissions

4) Please provide a detailed Financial Disclosure statement. This is published with the article. It must therefore be completed in full sentences and contain the exact wording you wish to be published.

1) Please clarify all sources of financial support for your study. List the grants, grant numbers, and organizations that funded your study, including funding received from your institution. Please note that suppliers of material support, including research materials, should be recognized in the Acknowledgements section rather than in the Financial Disclosure

2) State the initials, alongside each funding source, of each author to receive each grant. For example: "This work was supported by the National Institutes of Health (####### to AM; ###### to CJ) and the National Science Foundation (###### to AM)."

3) State what role the funders took in the study. If the funders had no role in your study, please state: "The funders had no role in study design, data collection and analysis, decision to publish, or preparation of the manuscript."

4) If any authors received a salary from any of your funders, please state which authors and which funders..

5) Your current Financial Disclosure states, "The author(s) received no specific funding for this work.".

However, your funding information on the submission form indicates receiving fund.

Please indicate by return email the full and correct funding information for your study and confirm the order in which funding contributions should appear. Please be sure to indicate whether the funders played any role in the study design, data collection and analysis, decision to publish, or preparation of the manuscript.

**Reviewers' comments:**

Reviewer's Responses to Questions

**Comments to the Authors:**

Reviewer #1: Deimler et al. describe a software pipeline (TARPON- Telomere Analysis and Research Pipeline Optimized for Nanopore) for computationally detecting and analyzing telomere sequence-containing DNA reads derived from Oxford Nanopore Technology (ONT) single-molecule long-read datasets. The pipeline has several important general advantages over other existing pipelines for analyzing telomere-containing nanopore reads, including ease of use combined with easily modifiable parameter modifications by non-bioinformaticists; modification of key analysis parameters through either GUI or command line interfaces and integration of all steps into a single Nextflow pipeline ensures reproducibility and ease of use. In addition, several important and useful innovations are incorporated into their pipeline; (1) a broader definition and systematic incorporation of (TTAGGG)n-like repeat(s) (which they refer to as GGATTG +1N) into subtelomere-telomere boundary definition, with their consequent incorporation into all downstream telomere terminal repeat analysis (telomere length determination, and potential (TTAGGG)n -like repeat composition and organization analyses); (2) automated detection and analysis of terminal telomere repeat tracts, telomere capture probes, and sequence tags used for multiplexing samples and (3) a set of thoughtfully designed and empirically tested filtering steps to help ensure that only reads with full-length terminal repeat regions are included for telomere length determination and other downstream analyses.

The logic, clarity, and detail for the results underlying Figures 1-5 (pre-mapping telomere read detection, filtering steps, and telomere length & composition analyses of telomere-containing single-reads) is for the most part commendable. Exceptions to this which require remediation are:

(A) Fig 1C, which has print content much smaller (about 10-fold ?) than the rest of the figures and is nearly impossible to read.

(B) The precise definition of GGATTG + 1N, which is ambiguous as written. Are 5-mer, 7-mer, and 6-mer single-base substitution variants included ? Does this set of variants comprehensively cover the known human (TTAGGG)n -like variants in terminal repeat tracts ? There should be sufficient high-quality telomere sequence data publicly available to actually check this, some effort should have been made to determine whether this variant set actually covers known variants, or if some are missed. A supplementary table/comprehensive list showing all variants used for this pipeline should be provided.

(C) Several overly sparse figure legends, and confusing figures which do not clearly describe the data presented. For example:

4b is unclear, plotting % repeats in the “first 300 bp of a read” for the four datasets – “first” from which end of the read and for which strand ? Please clarify in the figure legend. This criterion caused removal of lots of reads from the datasets (4c), so its critical to understand whats happening and to do this correctly.

4d-4h: The steps to define the subtelomere-telomere boundary in 4d-h should be moved to supplementary figures and described in detail there, including expanded figure legends and relevant text from the main paper. Only the final best conditions should be shown in the main part of the paper, since these are the ones defined for use; summary figures 4i and 4j should also be included in the main part of the paper here.

Fig 5a X-axis states “Single Nucleotide Repeat Composition [%]”. What does this mean ?

Fig 5g – What do the colors designate ?

Supplemental Figure 5 – Please explain in more detail what this is and how it was determined.

(D) The actual capture sequences used for the Splint-capture libraries comprising two of the source datasets analyzed in the paper were not provided. This makes it difficult to precisely replicate the entire pipeline.

A major issue is that the section of the paper describing the mapping and clustering of telomere reads (the results shown in Figure 6 and associated supplementary figures) does not provide sufficient detail to understand what was done and what the results are supposed to be showing. This single-telomere-read mapping and telomere read clustering section is fraught with major issues and uncertainties that are not addressed adequately.

Specifically:

E)

How exactly was the read-mapping done ? There is no methods section describing the exact algorithms and parameters used to acquire the results shown. I’m assuming some version of minimap2 was used, but insufficient discussion of the mapping parameter threshholds used for declaring a read “uniquely mapped” is given. For example, how is the Mapping Quality Score arrived at ? Will this vary with the basecaller, does it mean the same thing near telomeres as it does in less complex genome regions, and what are the variables contributing to this number ? These parameters as well as the actual nanopore read quality within subtelomere regions are expected to be critical for assessing the level of certainty that a read mapping is unique and correct. Especially for the relatively short distal stretches of subtelomeres that seemed to be the main focus of the read mapping and clustering analyses, there are highly similar hypervariable VNTRs as well as variable organizations of highly similar segmental duplication segments at many subtelomeres.

While the mapping process is treated as a black box, the data presentation seems to emphasize the positives. For example, while Figs 6a – 6c indeed suggest (as expected) that the telomere read mapping works best using a reference sequence source genome identical to that from which the telomere reads were derived, it doesn’t really address how accurate the single-telomere mapping specificity is for these 1-pass error-prone reads. The argument for telomere specificity of read-mappings is made in part by showing similar telomere read coverages at all chromosome ends in HG002 (6b); but it seems to be made after averaging the number of reads at all arms contributing to the lowest quartile of reads per arm, and using that number to normalize the mapping number per chromosome arm. Why not instead provide chromosome end by chromosome end raw read mapping numbers here, and let the reader dig in to these mapping data to investigate individual reads ? In Figure 6c, what is the average query alignment length as a fraction of the length of the subtelomeric part of the telomere-containing read (are there significantly sized subtelomeric segments of query reads not aligning or mis-aligning to the reference ? – and if so, how does this vary by chromosome end ?).

F) There are similar issues with the clustering of telomere reads by sequence similarity using Telogator2 (very little description of what Telogator2 is and how it works, another “black box” producing clusters of unknown quality/confidence from the nanopore telomere reads). From previously published work, Telogator2 seems very effective for clustering of telomeres using (TTAGGG)n-like repeat patterns in the proximal region of terminal repeat tracts sequenced with high-quality HiFi methods, but it worked poorly with nanopore reads base-called several years ago and it remains unclear how effectively it might work with enriched telomere libraries sequenced using current nanopore methods and current basecallers.

As with some of the other results, Figures 6d-g describing the clustering results were difficult to follow because of very sparse figure legends and confusing figure labels (eg., distance before telomere start, distance after telomere start, telomeric sequence %, telomere sequence #, telomeric sequence per cluster %, clusters #). No rationale is given for why telomere clusters are defined as containing >0.02% of the total number of input telomere reads. There is no descriptor or metric that I could ascertain amongst the results for measuring cluster quality, which would seem to me to be an extremely important parameter. It seems like HG002 might be an ideal model to develop Telogator quality metrics, as HG002 assembly used HiFi sequences extending into telomeres, and the current study includes telomere-enriched libraries sequenced from HG002 using current nanopore methods. As currently written, I cannot understand and don’t really trust the clustering results in Figure 6.

Reviewer #2: Paper Summary:

In this paper, the authors have developed a software called TARPON which can perform end-to-end chromosome arm-specific telomeric sequence and length analysis specialized for ONT reads. This is designed as a Nextflow pipeline that can either be executed via the command-line or the EPI2ME GUI. With respect to the existing telomere analysis tools like Telometer, TeloBP, and wf-teloseq, TARPON provides a better all-in-one experience for the users by taking over the data pre-processing steps within its integrated workflow. Unlike pre-existing tools, TARPON can serve as an easy-to-use tool for non-expert users with its default settings and also offers increased flexibility to the advanced users without needing to modify the source code.

Strengths:

- Well-written: The manuscript is easy to read and explains all the components in a comprehensive manner. The authors have included the limitations of the current version of the software. They have also suggested how the users can optimize the parameter values for different application scenarios.

- Meticulous methodology: TARPON’s workflow consists of novel components which are not present in the existing telomere analysis tools. The authors started with details of the lab protocols followed for sample preparation which ensures reproducibility. The authors later explained each step involved in the TARPON pipeline with adequate reasoning.

- Comprehensive experiments and result analysis: The authors performed thorough experimentation on diverse datasets and reported the results with detailed illustrations. The result analyses are sufficient to validate the performance of TARPON on human samples.

- High quality illustrations: The authors have performed thorough experimentations and corresponding analyses. The results are presented using high quality and easily interpretable figures.

- Strong motivation and applicability: The manuscript explains the shortcomings of TRF, qPCR, and FISH. It also explicates the benefits of utilizing ONT reads with enrichment techniques for telomere analysis with respect to cost and efficiency.

- Well-organized outputs: TARPON generates the outputs in publication-ready format including customizable statistics and summaries.

- Easy-to-follow instructions: The software installation instructions are well documented. The users don’t require in-depth computational knowledge to perform them.

Major Questions:

- Although the authors claim that TARPON is applicable to telomere analyses in variant-rich samples and organisms (some insects and plants) with non-canonical telomeric repeats, they don’t provide any experimental validation for this claim. Did the authors run experiments on those datasets, but omitted the results in the manuscript? In that case, it would be interesting to get a look at those results.

- It was intuitive that results obtained from the ONT reads and the corresponding reference from the same sample (HG002 in this case) would be the best. Likewise, when analysis is done between reads and reference coming from different samples would perform poorly. I am not sure what new information the readers would gain from this set of experimental analysis.

- For the tests on clinical samples, B2_duplex and B2_simplex, initially there were 2,034 and 3,170 telomeric reads. Later, it was mentioned that the number of telomeric reads were increased to 6,266, which improved the results. It was not clear how the authors increased the number of telomeric reads to 6,266. Did they use ONT reads generated with higher coverage depths or something else?

- How did the authors conclude that a minimum of 5,000 telomeric reads are required per sample for de novo chromosome arm-specific telomere length analysis?

Minor Comments:

- There are a few typos and inconsistencies. Before publication, the manuscript must go through thorough proof-reading. For example:

-- Page 15, Line 352-353: “... SUP basecalling basecalling (Fig. 3c).” basecalling is written twice.

-- Page 15, Line 363: “... after the identification of twenty telomeric repeats.” Previously, it was mentioned that ten telomeric repeats.

-- Page 20, Line 483: has an extra space here, “... identified .”

-- Figures 5b and 5f are missing “[” and “]” in their y-axis labels

-- BP and bp are used interchangeably; should be consistent

-- For referring to coverage, “X” and “x” are used interchangeably; should be consistent

-- Page 23, Line 548: should be “Supplemental Fig. 7a”

-- Page 31, Line 757: “... de novo telomeres reads …”; should be telomeric reads.

- It is not clear what the three plots in supplemental figure 2 are referring to. A better explanation is required for the ease of understanding.

- The authors should update the link to the software’s GitHub repository since the one mentioned in the manuscript is not being maintained any more.

- I tried to install TARPON in my macbook locally and execute the pipeline on the provided sample dataset following the instructions from the GitHub repository:

-- First, it failed to execute as the default # of CPUs were set to 10, but my machine has 8 cores. Can the source code be modified to automatically take the maximum number of CPUs available in the machine it is running on, if it is less than 10?

-- Next, I executed adding the “--threads” parameter and setting its value to 8. This time, it exited on another error related to the docker. I assume the software is not locally executable on macOS. If that is the case, it should be mentioned in the manuscript that the stand-alone version is platform-dependent.

Reviewer #3: In this manuscript, authors intended to show TARPON is a comprehensive and modular pipeline for Nanopore-based telomere analysis, offering high flexibility. While the default settings are optimized for telomerase-positive human samples, all parameters are easily adjustable via the GUI or command line. This flexibility allows TARPON to support the analysis of organisms with non-canonical telomeric repeats and variant repeat-rich samples.

However, TARPON clearly acknowledges its limitations.

1. Limitation for Heterogeneous Repeat Structures: TARPON is currently not recommended for organisms with highly heterogeneous sequence repeats. This is an honest admission that TARPON's core VRR definition methodology is designed to handle relatively uniform Telo+1N variations and is not suitable for complex, heterogeneous repeat structures.

2. Clinical Validation Status: The manuscript explicitly states that neither Nanopore sequencing technology nor the TARPON pipeline has been formally clinically validated and should not be used as the sole clinical diagnostic methodology at this time. This disclaimer is important for maintaining scientific rigor and promoting responsible technology usage.

These limitations clearly define TARPON's current scope and emphasize the need for further research towards clinical adoption.

There are some recommendations to be addressed for the improvement of the manuscript

1. Although elaboration on the comparison among different existing telomere analysis methods can be found, a formal figure seems to be necessary to address the differences in actual data output to provide more information regarding the benchmarking process.

2. The wf-teloseq MAE of 42nt must be removed. Instead, TARPON (4.36nt) should be compared to wf-teloseq's raw MAE (157nt), and the reasons for wf-teloseq's susceptibility to misidentifying subtelomeric islands (algorithmic flaw) should be described in a separate paragraph. The post-filtered comparison result is not statistically justifiable.

3. The manuscript must explicitly acknowledge that including the VRR region in telomere length measurement causes a systematic length difference (overestimation) compared to traditional canonical repeat-based measurement methods, and quantitative data on this difference should be presented to assist readers in interpreting the measurement results.

4. The more accurate term "Telomere Allele Specific Telomere Length" must be consistently used throughout the manuscript instead of "Chromosome Arm-Specific Telomere Length" when describing de novo clustering results.

5. The high proximal truncation read removal rate (∼50%) observed in Duplex-enriched libraries (especially HEK-DE) should be highlighted as an experimental limitation of the Duplex Capture protocol. Users should be warned that this method may yield a lower proportion of usable full-length reads compared to the Splint-enriched method.

6. Final confirmation should be made that the putative telomere sequence UBAM files for the four samples used in the study will be made publicly available on SRA at the time of manuscript publication, as stated in the Data Availability Statement. Adherence to this commitment is crucial for ensuring scientific transparency.

7. It is unclear about which basecaller method (Fast/SUP/hybruid format, etc.) was used in the later part of the analysis (corresponding to the results in Fig.3 and on) inside the manuscript. It would be great to include such information inside the manuscript.

8. A figure representing the TARPON method (+ other methods like TeloBP / described in the discussion section of the manuscript) regarding identification of the telomeric/subtelomeric region boundary would be substantially beneficial (since this is critical to not only define the telomere sequence, but also the telomere length, which is imperative in cancer research area).

**Have the authors made all data and (if applicable) computational code underlying the findings in their manuscript fully available?**

Reviewer #1: Yes

Reviewer #2: Yes

Reviewer #3: Yes

PLOS authors have the option to publish the peer review history of their article (what does this mean?). If published, this will include your full peer review and any attached files.

Reviewer #1: No

Reviewer #2: **Yes:** Sakshar Chakravarty

Reviewer #3: No

**Figure resubmission:**
---

## [Decision Letter · Decision Letter 1]

12 Jan 2026

Dear Mr. Deimler,

We are pleased to inform you that your manuscript 'TARPON - a Telomere Analysis and Research Pipeline Optimized for Nanopore' has been provisionally accepted for publication in PLOS Computational Biology.

Best regards,

Adam Ewing

Academic Editor

PLOS Computational Biology

Shaun Mahony

Section Editor

PLOS Computational Biology

Reviewer's Responses to Questions

**Comments to the Authors:**

Reviewer #2: I thank the authors for addressing all of my concerns, which has resulted in a clear improvement in the quality and readability of the manuscript. I will review the GitHub repository again and will report an issue if the previously encountered problem persists.

Reviewer #3: I believe the authors have adequately addressed most of the concerns and made appropriate revisions.

**Have the authors made all data and (if applicable) computational code underlying the findings in their manuscript fully available?**

Reviewer #2: Yes

Reviewer #3: Yes

PLOS authors have the option to publish the peer review history of their article (what does this mean?). If published, this will include your full peer review and any attached files.

Reviewer #2: **Yes:** Sakshar Chakravarty

Reviewer #3: No

---

## [Editor Report · Acceptance letter]

PCOMPBIOL-D-25-01744R1

TARPON - a Telomere Analysis and Research Pipeline Optimized for Nanopore

Dear Dr Deimler,

I am pleased to inform you that your manuscript has been formally accepted for publication in PLOS Computational Biology. Your manuscript is now with our production department and you will be notified of the publication date in due course.

With kind regards,

Anita Estes
